# Salmonella enters a dormant state within human epithelial cells for persistent infection

Chak Hon Luk[1,2], Camila Valenzuela[1º], Magdalena Gil[1º], Léa Swistak[1,2º], Perrine Bomme[3], Yuen-Yan Chang[1], Adeline Mallet[3], Jost Enninga[1,2]*

1 Dynamics of Host-Pathogen Interactions Unit and UMR3691 CNRS, Institut Pasteur, Paris, France, 2 Université de Paris, Sorbonne Paris Cité, Paris, France, 3 Ultrastructural Bioimaging UTechS, C2RT, Institut Pasteur, Paris, France

º These authors contributed equally to this work.
* jost.enninga@pasteur.fr

**Data Availability Statement:** All relevant data are within the manuscript and its Supporting Information files.

## Abstract

*Salmonella* Typhimurium (*S.* Typhimurium) is an enteric bacterium capable of invading a wide range of hosts, including rodents and humans. It targets different host cell types showing different intracellular lifestyles. *S.* Typhimurium colonizes different intracellular niches and is able to either actively divide at various rates or remain dormant to persist. A comprehensive tool to determine these distinct *S.* Typhimurium lifestyles remains lacking. Here we developed a novel fluorescent reporter, *Salmonella* INtracellular Analyzer (SINA), compatible for fluorescence microscopy and flow cytometry in single-bacterium level quantification. This identified a *S.* Typhimurium subpopulation in infected epithelial cells that exhibits a unique phenotype in comparison to the previously documented vacuolar or cytosolic *S.* Typhimurium. This subpopulation entered a dormant state in a vesicular compartment distinct from the conventional *Salmonella*-containing vacuoles (SCV) as well as the previously reported niche of dormant *S.* Typhimurium in macrophages. The dormant *S.* Typhimurium inside enterocytes were viable and expressed *Salmonella* Pathogenicity Island 2 (SPI-2) virulence factors at later time points. We found that the formation of these dormant *S.* Typhimurium is not triggered by the loss of SPI-2 effector secretion but it is regulated by (p) ppGpp-mediated stringent response through RelA and SpoT. We predict that intraepithelial dormant *S.* Typhimurium represents an important pathogen niche and provides an alternative strategy for *S.* Typhimurium pathogenicity and its persistence.

## Author summary

*Salmonella* Typhimurium is a clinically relevant bacterial pathogen that causes Salmonellosis. It can actively or passively invade various host cell types and reside in a *Salmonella*-containing vacuole (SCV) within host cells. The SCV can be remodeled into a replicative niche with the aid of *Salmonella* Type III Secretion System 2 (T3SS2) effectors or else, the SCV is ruptured for the access of the nutrient-rich host cytosol. Depending on the infected host cell type, *S.* Typhimurium undertake different lifestyles that are distinct by their subcellular localization, replication rate and metabolic rate. We present here a novel

**Funding:** This research was supported by fellowships from Croucher Foundation (HK) and Fondation pour la Recherche Médicale (FRM) to C. H.L. and Y.Y.C.. C.H.L. is part of the Pasteur - Paris University (PPU) International PhD Program. J.E. is supported by the ERC-CoG "Endosubvert". The Enninga lab is part of the LabEx IBEID and Milieu Interieure. AM and PB are supported for equipment from the French Government Programme Investissements d'Avenir France BioImaging (FBI, N° ANR-10-INSB-04-01) and are also members of the LabEx IBEID. The funders had no role in study design, data collection and analysis, decision to publish, or preparation of the manuscript.

**Competing interests:** The authors have declared that no competing interests exist.

fluorescent reporter system that rapidly detects *S*. Typhimurium lifestyles using fluorescence microscopy and flow cytometry. We identified a dormant *S*. Typhimurium population within enterocyte that displays capacities in host cell persistence, dormancy exit and antibiotic tolerance. We deciphered the (p)ppGpp stringent response pathway that suppresses *S*. Typhimurium dormancy in enterocytes while promoting dormancy in macrophages, pinpointing a divergent physiological consequence regulated by the same set of *S*. Typhimurium molecular mediators. Altogether, our work demonstrated the potential of fluorescent reporters in facile bacterial characterization, and revealed a dormant *S*. Typhimurium population in human enterocytes that are phenotypically distinct from that observed in macrophages and fibroblasts.

## Introduction

*Salmonella enterica* serovar Typhimurium (*S*. Typhimurium) is an enteric bacterium that closely associates with global food-borne illnesses. The prevalence of *S*. Typhimurium has placed a severe burden on the global food and healthcare industry, leading to millions of cases, hundreds of casualties and costing billions of dollars per annum [1,2]. *S*. Typhimurium resides in different natural reservoirs and is transmitted to humans through contaminated food.

Upon arrival in the human intestine after ingestion, a portion of the luminal *S*. Typhimurium expresses the Type III Secretion System 1 (T3SS1) encoded within *Salmonella* Pathogenicity Island 1 (SPI-1) and its cognate effectors to induce its active entry into non-phagocytic epithelial cells. *S*. Typhimurium also targets other cell types, such as fibroblasts and macrophages. During these events, it induces local tissue injuries and eventually breaches the intestinal barrier to reach the lamina propria and tissue-resident immune cells. Then, *S*. Typhimurium is carried by macrophages to mesenteric lymph nodes and eventually to the liver and spleen for persistent infection [3].

Within enterocytes *S*. Typhimurium are encapsulated in an endocytic compartment coined *Salmonella*-containing vacuole (SCV) that matures by acidification within the first hours of internalization [4]. The reducing pH and changing osmolarity of the SCV induce the shutdown of T3SS1 and expression of a second T3SS, T3SS2, from *Salmonella* Pathogenicity Island 2 (SPI-2) [5,6]. The T3SS2 effectors remodel the SCV into a viable niche for *S*. Typhimurium replication [5,6]. Default maturation of the SCV is marked by the sequential acquisition and removal of endocytic trafficking markers, such as the small GTPases RAB5 and RAB7 as well as Lysosome-associated membrane glycoprotein 1 (LAMP1) [7]. During these events, the SCV dynamically interacts with surrounding macropinosomes, which controls SCV stability [8,9]. Consequently, *S*. Typhimurium can reside either in a remodeled SCV or disrupt the SCV to access the host cytosol. Overall, *S*. Typhimurium exhibits distinct replication rates and specific metabolic profiles within the different intracellular niches adapting to nutrient availability and the specific microenvironments [8–10].

Differential lifestyles are also known for *S*. Typhimurium infecting other cell types. In fibroblasts, the SCV associates with the aggrephagy machinery that either clears the infection or allows *S*. Typhimurium to putatively persist in the cell [11,12]. In macrophages, *S*. Typhimurium expresses the T3SS2 to remodel the SCV immediately after bacterial entry, or the pathogen adopts a dormant behavior mediated by toxin-antitoxin (TA) system [13,14].

Antibiotic persistence and relapse of *S*. Typhimurium infection due to the failure of bacterial eradication through antibiotic treatment has been tied to *S*. Typhimurium dormancy, which is distinct from bacterial persistence that refers to incompletely cleared infections by the

immune system. Numerous antibiotics target major active machineries, including DNA replication, transcription and translation of extracellular bacteria, therefore dormant intracellular pathogens appear to be less or not susceptible to such treatments [15]. *S.* Typhimurium dormancy and antibiotics persistence have been reported in macrophages to be regulated by the Guanosine pentaphosphate (ppGpp) stringent response pathway. The two (p)ppGpp synthases RelA and SpoT control the bacterial (p)ppGpp level, which regulates the activity of the ATP-dependent protease Lon to degrade the antitoxin and release the toxin TacT for arresting protein translation [14,16]. The arrest of translation by TacT leads to a halt of bacterial growth giving rise to the insensitivity and tolerance towards antibiotics [16]. Despite reports of *S.* Typhimurium antibiotics persistence in the epithelium and lamina propria of the mouse intestine, it is not clear whether this involves dormant bacteria [17].

With our *Salmonella* INtracellular Analyzer (SINA) system, we precisely depicted the intracellular bacterial lifestyles at the single bacterium level, identifying a novel *S.* Typhimurium population within enterocytes that is dormant. Dormant persisters within enterocytes are localized in a unique vacuolar compartment different from the one described in macrophages. We found that T3SS2 effector secretion and the Lon protease are dispensable for this new *S.* Typhimurium population, while the bifunctional enzyme SpoT and monofunctional enzyme RelA negatively regulate *S.* Typhimurium dormancy in epithelial cells.

## Results

### Development of a multiplex fluorescent reporter series, the *Salmonella* INtracellular Analyser (SINA) to distinguish different intracellular *S.* Typhimurium lifestyles

The distinct intracellular lifestyles of *S.* Typhimurium upon invasion of epithelial cells have been described either with regard to the specific pathogen localization or with regard to the bacterial growth dynamics. To date, fluorescent reporters are available to identify *S.* Typhimurium within vacuolar and cytosolic localizations; while the others measure the replication rate of the pathogen [18–20]. However, these localization and replication-rate reporters have not been coupled, as it has been generally assumed that the bacterial localization determines its replication rate. This notion has been challenged by different reports, for example on the different growth rates of *S.* Typhimurium within the cytosol depending on the targeting by autophagy [21–25]. A combined reporter system would enable a comprehensive elucidation of the intracellular lifestyle of a given intracellular pathogen. Therefore, we developed a novel fluorescent reporter series, the *Salmonella* INtracellular Analyzer (SINA). Our SINA1.1 reporter is composed of two separated modules to indicate the bacterial localization and replication rate. The localization module consists of two transcription reporters driven by localization-specific promoters, while the replication rate module carries a constitutively expressed fluorescent timer (Figs 1A and S1). At the molecular level, the localization module is composed of vacuolar (Vac) and cytosolic (Cyt) submodules, which utilize two characterized promoters, $P_{ssaG}$ and $P_{uhpT}$ to drive the expression of tagBFP and smURFP, respectively [18,19]. We confirmed the functionality of the fluorescent $P_{ssaG}$ and $P_{uhpT}$ reporters for our experimental setup during *S.* Typhimurium invasion of epithelial cells using a digitonin assay assayed by flow cytometry (S2 and S3 Figs). The replication rate module encodes Timer[bac], a DsRed mutant (S197T), which has been previously employed to differentiate *S.* Typhimurium subpopulations by their replication rates [20]. The emission spectrum of Timer[bac] shifts from green to red as it matures, which reflects the bacterial metabolic activity (change in slope, Fig 1B top) as well as the replication rate (unvarying slope, varying green:red ratio, Fig 1B bottom) [26]. When Timer[bac] is constitutively expressed, a metabolically active *S.* Typhimurium bacterium emits both green

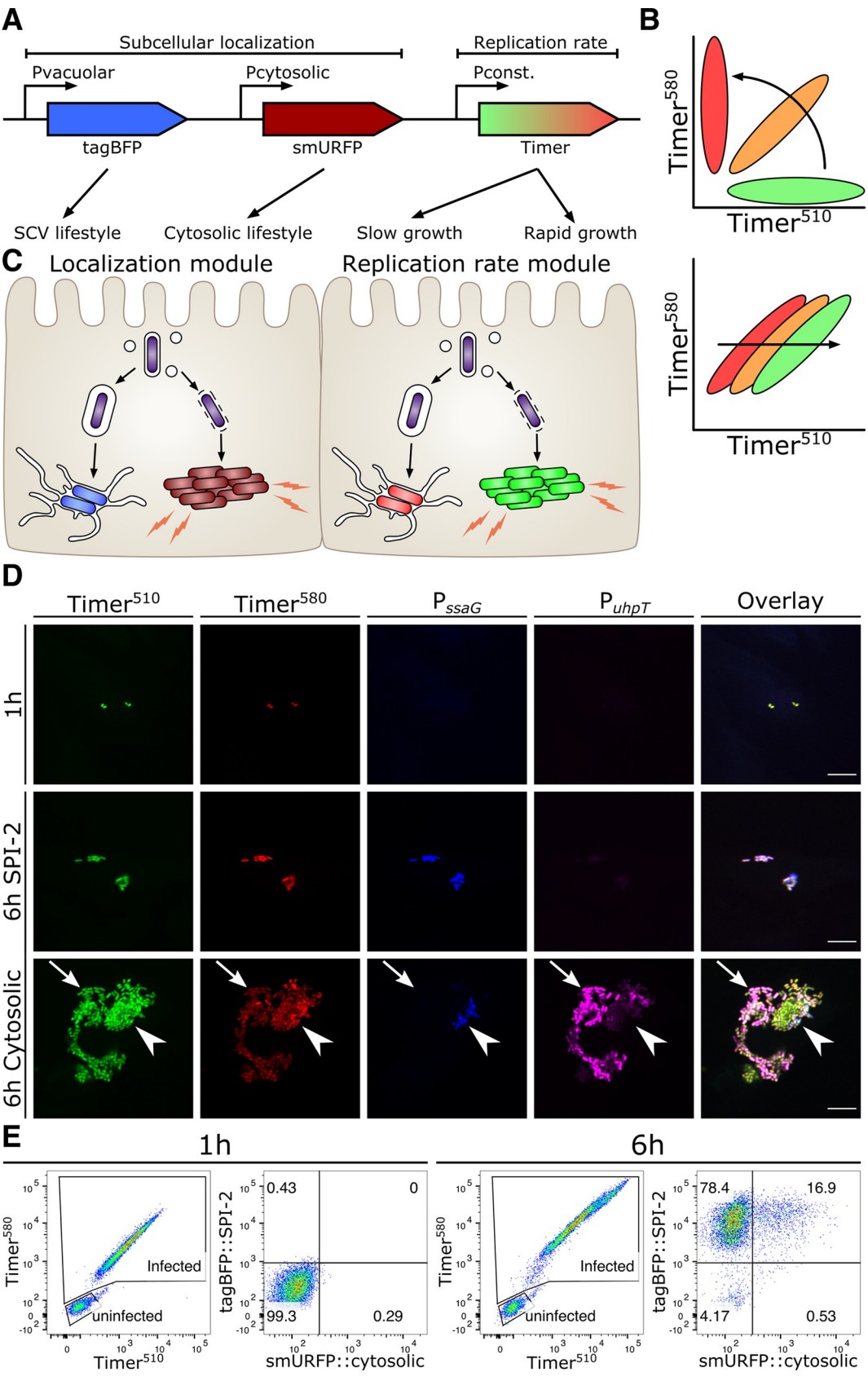

**Fig 1. SINA enables precise determination of the different *Salmonella* intracellular lifestyles in human epithelial cells.**
(A) Schematic diagram of the construction of subcellular localization and replication rate modules of SINA1.1. The subcellular localization module is composed of the vacuolar submodule ($P_{ssaG}$-tagBFP) and cytosolic submodule ($P_{uhpT}$-smURFP), while the replication rate module is composed of a constitutively expressed Timer$^{bac}$ ($P_{ybaJ}$-Timer$^{bac}$) (B) (Top) Schematic diagram of the emission spectrum shifts of *S.* Typhimurium harboring Timer$^{bac}$ as Timer$^{bac}$ matures, where emission shifts from green to red (Bottom) Green:Red ratio increases with elevating *S.* Typhimurium replication rates. As *S.* Typhimurium divides, both Timer$^{510}$ and Timer$^{580}$ fluorophores are diluted. With a higher production rate of Timer$^{510}$ than Timer$^{580}$, fast dividing *S.* Typhimurium exhibits a higher Green:Red ratio. (C) Expected output by SINA as *S.* Typhimurium dwells in distinct subcellular localizations. Vacuolar *S.* Typhimurium are of lower replication rate (i.e. lower Green:Red ratio) and are expected to emit blue fluorescence; cytosolic *S.* Typhimurium are of higher replication rate (i.e. higher Green:Red ratio) and are expected to emit far red fluorescence (D) HeLa cells infected by *S.* Typhimurium harboring SINA1.1. Output of SINA from intracellular *S.* Typhimurium was detected by fluorescence microscopy at 1 h pi, vacuolar (arrowhead) and cytosolic (arrow) *S.* Typhimurium at 6 h pi. (3 independent experiments). Scale bars are 10 μm. (E) HeLa cells infected by *S.* Typhimurium harboring SINA1.1. Output of SINA from intracellular *S.* Typhimurium at 1 h and 6 h pi was detected by flow cytometry (3 independent experiments).

and red signals resulting from immature green and mature red fluorophores. In case such a bacterium experiences a metabolic halt, it eventually emits only red signals, due to the maturation of the existing green fluorophores in concert with ceased *de novo* synthesis of the green fluorophores. With SINA, we are able to simultaneously collect information on these replication rate changes of *S.* Typhimurium and its localization at single bacterium resolution, which enables a comprehensive and quantitative reflection of *S.* Typhimurium physiology inside an infected host.

To validate the functionality of our SINA system during *S.* Typhimurium invasion of epithelial cells, we employed fluorescence microscopy and flow cytometry analysis (Fig 1C). With fluorescence microscopy, we observed intracellular *S.* Typhimurium simultaneously emitting both green and red signals (Timer$^{bac}$), but not Vac ($P_{ssaG}$-*tagBFP*) and Cyt ($P_{uhpT}$-*smURFP*) signals at 1 hour post-infection (pi) (Fig 1D). As these bacteria committed to vacuolar or cytosolic lifestyles at 6 hours pi, we observed that *S.* Typhimurium in cells with <10 bacteria emitted Vac signal during this time course. On the other hand, we observed a mixed population of *S.* Typhimurium in cells with >10 bacteria, where clusters of Cyt$^+$ *S.* Typhimurium of low Timer$^{bac}$ signals (arrow) and individual Vac$^+$ *S.* Typhimurium (arrowhead) were detected. In the cells containing mixed *S.* Typhimurium populations, bacteria were either Vac$^+$ or Cyt$^+$ but not double positive, showing the presence of two populations with distinct discernible lifestyles (Fig 1D). We were also able to track the onset of bacterial division and signal output from SINA by time-lapse microscopy (S1–S3 Movies, S4B and S4C Fig).

We took advantage of the properties of our multiplex SINA reporter and devised a gating strategy to quantitatively analyze the bacterial lifestyles in single *S.* Typhimurium-infected cells using flow cytometry (S2 Fig). In brief, we first defined the infected cells by the size of the analyzed events (under SSC-A vs FSC-A plot), followed by the positive signals in the Timer$^{580}$ vs Timer$^{510}$ plot (i.e. cells harboring *S.* Typhimurium). We then further classified the *S.* Typhimurium-infected cells into four sub-types according the signals of the localization module (tagBFP::SPI-2 vs smURFP::cytosolic plot), corresponding to cells with either vacuolar bacteria (Vac$^+$Cyt$^-$) or cytosolic bacteria (Vac$^-$Cyt$^+$) or cells with both vacuolar and cytosolic populations (Vac$^+$Cyt$^+$) or cells harboring *S.* Typhimurium that express only basal levels of the Vac and Cyt signals (S2 Fig). We observed that intracellular *S.* Typhimurium behaved as a population with a homogenous replication rate and basal expression levels of Vac and Cyt at 1 hour pi (Fig 1E). At 6 hours pi, this homogenous population segregated into Vac$^+$ and Cyt$^+$ subpopulations, with a Cyt$^+$ distribution similar to that reported in the literature (10–20% cytosolic) (Fig 1E) [21]. The gradual separation of these subpopulations could be detected with SINA1.1 throughout the course of infection (S4A Fig). As we backgated the infected cells, we observed cells harboring only vacuolar *S.* Typhimurium (Vac$^+$Cyt$^-$) and cells with both vacuolar and

cytosolic bacteria (Vac$^+$Cyt$^+$), each forming a distinct population of different Green:Red ratio on the plot of Timer$^{510}$ against Timer$^{580}$ (S5A Fig). Together, this demonstrated that our novel SINA1.1 reporter is capable of simultaneously and quantitatively distinguishing the *S.* Typhimurium lifestyles by their subcellular localization and replication rate at both single infected cell and single bacterium level using flow cytometry and fluorescence microscopy. The combination of the SINA reporter with flow cytometry fosters higher throughput analysis of *S.* Typhimurium lifestyles in infected cells as compared to microscopy, extending the possibility for rapid screening.

## A novel dormant *S.* Typhimurium subpopulation in human epithelial cell

With the SINA1.1 reporter, we used the SPI-2 expression module to distinguish vacuolar *S.* Typhimurium from cytosolic bacteria. In the plot of the localization module, we identified an easily discernable population (~5–10%) of infected epithelial cells harboring Vac$^-$Cyt$^-$ *S.* Typhimurium detectable as early as 2 hours pi, which became apparent at 6 hours pi (Figs 2A and S6). We backgated the Vac$^-$Cyt$^-$ population, and we extracted physical parameters from the Timer$^{bac}$ plot. This revealed that the Vac$^-$Cyt$^-$ *S.* Typhimurium exhibit a similar replication rate (S5A Fig) but a reduced metabolic activity (S5B Fig) compared to Vac$^+$Cyt$^-$ *S.* Typhimurium as depicted by the green:red ratio and slope of Timer$^{bac}$ plot, respectively. The capacities of Timer$^{bac}$ in the measurement of the bacterial replication rate and metabolism have been well-elaborated in previous applications [20,27]. This Vac$^-$Cyt$^-$ *S.* Typhimurium population was also visualized using live microscopy to confirm their presence using different detection approaches (S3 Movie). This was intriguing as metabolically inactive *S.* Typhimurium have not been reported in enterocytes so far. We thus infected polarized intestinal epithelial Caco-2 monolayers, and confirmed the presence of the Vac$^-$Cyt$^-$ subpopulation with a shifted metabolic profile in a cellular model system for intestinal infections (S7 Fig). We also performed control infections in 3T3 (fibroblast model) and differentiated THP-1 cells (macrophage model) to test the sensitivity of SINA1.1 in these relevant cell types. As within epithelial cells, we also observed distinct *S.* Typhimurium populations in the macrophage and fibroblast models as described before (S8 and S9 Figs). To determine the intracellular localization of Vac$^-$Cyt$^-$ *S.* Typhimurium, we further performed correlative light and electron microscopy (CLEM) by serial section transmission electron microscopy (TEM), and a digitonin assay demonstrating that this subpopulation is localized in a host vesicular compartment (Figs 2B, S4A–S4F and S10) [21]. Together, these results showed the presence of a novel intracellular *S.* Typhimurium population within epithelial cells that exhibits a lowered metabolic rate and resides in a host vesicular compartment, implicating a putative dormant phenotypic variant.

Intracellular *S.* Typhimurium encounters a number of stresses upon uptake into host cells, including oxidative, pH and osmotic stress, which serve as key signals to trigger transcription reprogramming for the adaptation of an intra-host environment [28]. As the intracellular microenvironment and *S.* Typhimurium dormancy has been studied in some detail in macrophages, we decided to focus our comparison on this cell type in relation to the newly identified *S.* Typhimurium subpopulation in epithelial cells. During macrophage infections the SCV microenvironment drives a portion of *S.* Typhimurium into a dormant state that contributes to the elevation of antimicrobial persistence and polarization of infected macrophage [14,29]. We asked whether Vac$^-$Cyt$^-$ *S.* Typhimurium shares similar physiologies with dormant *S.* Typhimurium inside macrophages, hence we determined the metabolic state of the Vac$^-$Cyt$^-$ population. By replacing the cytosolic submodule of SINA1.1 with an arabinose inducible cassette to generate SINA1.5, we measured *S.* Typhimurium's capacity to respond to arabinose treatment. This modification enabled us to directly monitor the metabolic activity of *S.*

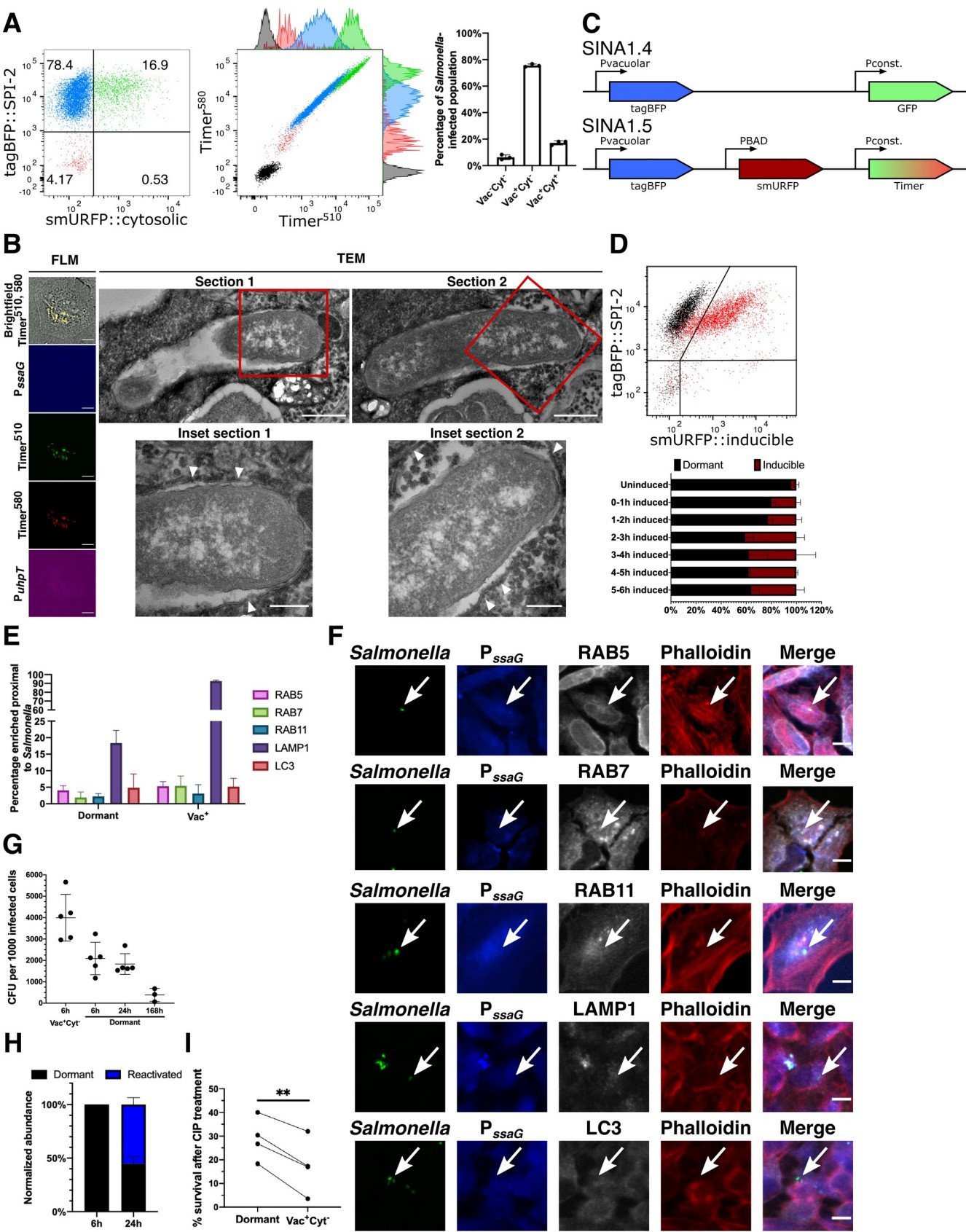

**Fig 2. *S*. Typhimurium displays a novel inactive intracellular lifestyle in epithelial cells.** (A) (Left and Middle) Timer^bac profile and distribution of single cells with no infection (black), infected cells with inactive bacteria (Vac⁻Cyt⁻) (red), infected cells with only vacuolar bacteria (Vac⁺Cyt⁻) (blue) and infected cells with both vacuolar and cytosolic populations (Vac⁺Cyt⁺) (green) at 6 h pi. (Right) Abundance of *S*. Typhimurium-infected cells (Vac⁻Cyt⁻, Vac⁺Cyt⁻ and Vac⁺Cyt⁺) as illustrated in (A) (n = 3). (B) (Left) Brightfield and fluorescent microcopy (FLM) images of infected HeLa cells harboring Vac⁻Cyt⁻ *S*. Typhimurium at 6 h pi. (Right) Serial sections of TEM images of Vac⁻Cyt⁻ *S*. Typhimurium, arrowhead indicates host membrane structures of the SCV. (C) Schematic illustration for the constructions of SINA derivatives, SINA1.4 and SINA1.5. SINA1.4 was used for immunofluorescence staining against RAB5, RAB7, RAB11, LAMP1 and LC3; SINA1.5 was used for arabinose induction assay. (D) (Top) Responsiveness of intracellular *S*. Typhimurium towards an arabinose pulse between 5–6 h pi, uninduced control (black); arabinose-induced (red). (Bottom) Quantification on the responsiveness of Vac⁻ *S*. Typhimurium pulsed at different time intervals during the infection time course, dormant (black), inducible (maroon). Samples were all harvested at 6 h pi. (n = 3) (E) HeLa cells were infected with SINA1.4-harboring *S*. Typhimurium, harvested at 6 h pi, fixed and stained. Quantification of the presence of RAB5, RAB7, BAB11, LAMP1 and LC3 proximal to Vac⁻ and Vac⁺ *S*. Typhimurium at 6 h pi. (n = 3) (F) Representative images of Vac⁻ *S*. Typhimurium (arrow) quantified in (D); *S*. Typhimurium (green), Vac⁻ (blue), RAB5, RAB7, RAB11, LAMP1 and LC3 (grey), Phalloidin (red). (G) Designated populations of infected HeLa cells were enriched by cell sorting and plated for CFU. Quantification of CFU from dormant *S*. Typhimurium at 6 h, 24 h and 168 h pi and Vac⁺Cyt⁻ *S*. Typhimurium at 6 h pi. (n = 5 for Vac⁺Cyt⁻ 6 h, Dormant 6 h, 24 h; n = 3 for Dormant 168 h) (H) Quantification of SPI-2 activity using flow cytometry in enriched dormant *S*. Typhimurium at 6 h and enriched Vac⁻Cyt⁻ infected cells re-plated until 24 h pi. (n = 3) (I) Survival percentage of dormant and Vac⁺Cyt⁻ intracellular *S*. Typhimurium against 3 h of CIP treatment, infected cells were harvested at 6 h pi, enriched by cell sorting and plated for CFU. (n = 4) At least a total of 1000 events of infected cells were analyzed by flow cytometry or 50 infected cells by microscopy in each experiment replicates. The bars represent the mean, statistics were performed using unpaired t test (**p<0.01).

Typhimurium during the infection process (Fig 2C). The response of intracellular bacteria towards extracellular arabinose induction has been reported previously to characterize the metabolic state of macrophage-borne dormant *S*. Typhimurium [13]. With SINA1.5, we observed that approximately half of the Vac⁻ *S*. Typhimurium did not respond to arabinose induction at designated time intervals (Fig 2D). As the Vac⁻Cyt⁻ bacterial niche may show limited arabinose accessibility, we corroborated our observations monitoring the reduced metabolism of this bacterial population via the signals of the Timer^bac reporter (S5B Fig). These data allowed us to propose that the Vac⁻Cyt⁻ *S*. Typhimurium adopts a dormant state (coined as dormant *S*. Typhimurium hereafter) upon their internalization into epithelial cells.

## Dormant *S*. Typhimurium resides in a unique vesicular compartment

We set out to determine whether the dormant *S*. Typhimurium localization is distinct from conventional SCVs. To determine this by immunofluorescence staining, we simplified SINA1.1 to SINA1.4 to free the red and far-red channels for indirect immunofluorescence staining of selected endocytic markers (Fig 2C). LAMP1 labels host lysosomes as well as the matured SCV, which is also present on the SCV of dormant *S*. Typhimurium in macrophages [7,13]. By fluorescence microscopy, we only observed minor recruitment of LAMP1 to the proximity of dormant *S*. Typhimurium within epithelial cells, in contrast to the high LAMP1 incidence proximal to Vac⁺ *S*. Typhimurium (Fig 2E and 2F). We also determined if dormant *S*. Typhimurium is localized in a SCV experiencing a halt in SCV biogenesis, therefore we tested a number of known early SCV markers (namely RAB5, RAB7 and RAB11). We did not observe the recruitment of any of these early SCV markers to the dormant *S*. Typhimurium (Fig 2E and 2F). This was intriguing as our previous ultrastructural and digitonin investigations clearly documented the dormant *S*. Typhimurium bacteria within a membrane-bound compartment. We also addressed if the dormant population is targeted by host autophagy, analyzing the localization of the autophagy marker LC3. We did not detect any localization of LC3 proximal to the majority of the dormant *S*. Typhimurium (Fig 2E and 2F). Therefore, we conclude that dormant *S*. Typhimurium are localized within a unique membrane-bound compartment distinct from the conventional SCV and that of dormant *S*. Typhimurium in macrophages, suggesting such dormancy formation is heavily governed by endocytic trafficking [13,30]. This compartment requires further characterization in future studies.

## Dormant *S*. Typhimurium are viable, cultivable, resume metabolism and express virulence genes in host cells

Endocytic vesicles are either recycled or undergo fusion with the lysosomes for degradation. The same fate also applies to vacuolar *S*. Typhimurium, where SPI-2 deficient strains have a reduced survival capacity compared to SPI-2 competent strains [31]. We collected the infected cells harboring dormant *S*. Typhimurium by cell sorting at >90% purity and plated them for colony forming unit (CFU) measurement, a classical approach to determine the viability of the intracellular *S*. Typhimurium. We observed that dormant *S*. Typhimurium are viable and cultivable (Fig 2G), contrasting to the viable but not cultivable nature of dormant *S*. Typhimurium in murine macrophages [13,14]. To determine the fate of the dormant *S*. Typhimurium, we enriched and plated the viable infected cells harboring dormant *S*. Typhimurium, and monitored the bacterial behavior at 24 hours pi (S11 Fig). We observed that 50% of the dormant *S*. Typhimurium in infected cells collected at 6 hours pi became metabolically active and expressed SPI-2 at 24 hours pi, as demonstrated by the population shift in the Timer$^{bac}$ plot, and becoming Vac$^+$ (Fig 2H). To determine if the dormant *S*. Typhimurium persists in the host, we further enriched infected cells harboring dormant *S*. Typhimurium and monitored the presence and the viability of dormant *S*. Typhimurium at 7 days pi. The dormant *S*. Typhimurium were found to persist in cells and remained viable and cultivable over the whole period of 7 days (Fig 2G, "168h"). To further address whether the dormant intracellular *S*. Typhimurium phenotype confers a reduced sensitivity towards antibiotics, we supplemented ciprofloxacin (CIP) to the infected cell at 3 hours pi and determined the viability of dormant *S*. Typhimurium by CFU. We selected the *in cellulo* CIP supplementation to avoid artificial perturbations on the *S*. Typhimurium dormant status. Also, *in cellulo* treatment addresses the killing efficiency of dormant *S*. Typhimurium within the SCV, where subcellular distribution of this drug has been demonstrated to influence the bactericidal efficacy [32]. We observed a higher survival rate of dormant *S*. Typhimurium as compared to vacuolar *S*. Typhimurium (Fig 2I), similar to the observations made in the murine intestine [17]. These results demonstrated that dormant *S*. Typhimurium are viable, exhibit a delayed expression of SPI-2, persist in the epithelial host cells for up to 7 days and are less susceptible to antibiotics. Such unique metabolic and virulence reprogramming could serve as a strategic step for intestine-borne *S*. Typhimurium to prolong gut inflammation for community benefits and reservoir for relapse [33].

## *S*. Typhimurium dormancy is not a result of the loss of T3SS2 effector secretion

We then studied whether the lack of T3SS2 effector secretion drives *S*. Typhimurium dormancy in epithelial cells. Using our SINA1.1 reporter in the *S*. Typhimurium SPI-2 secretion deficient mutant *ΔssaV* [34], we observed no significant difference in the proportion of Vac$^-$ Cyt$^-$ *S*. Typhimurium between wild type and the *ΔssaV* mutant (Fig 3A). Altogether, the formation of dormant *S*. Typhimurium is not a consequence caused by the lack of T3SS2 effector secretion during the infection of epithelial cells.

## *S*. Typhimurium dormancy is regulated by (p)ppGpp biogenesis

Class II toxin-antitoxin (TA) systems regulate the dormancy formation of non-pathogenic *E. coli* in laboratory conditions [35]. TA systems are comprised of a toxin and an antitoxin that counter-balances the toxin to regulate bacterial physiology, including growth arrest. A major TA system involves the stringent response mediated by the monofunctional (p)ppGpp

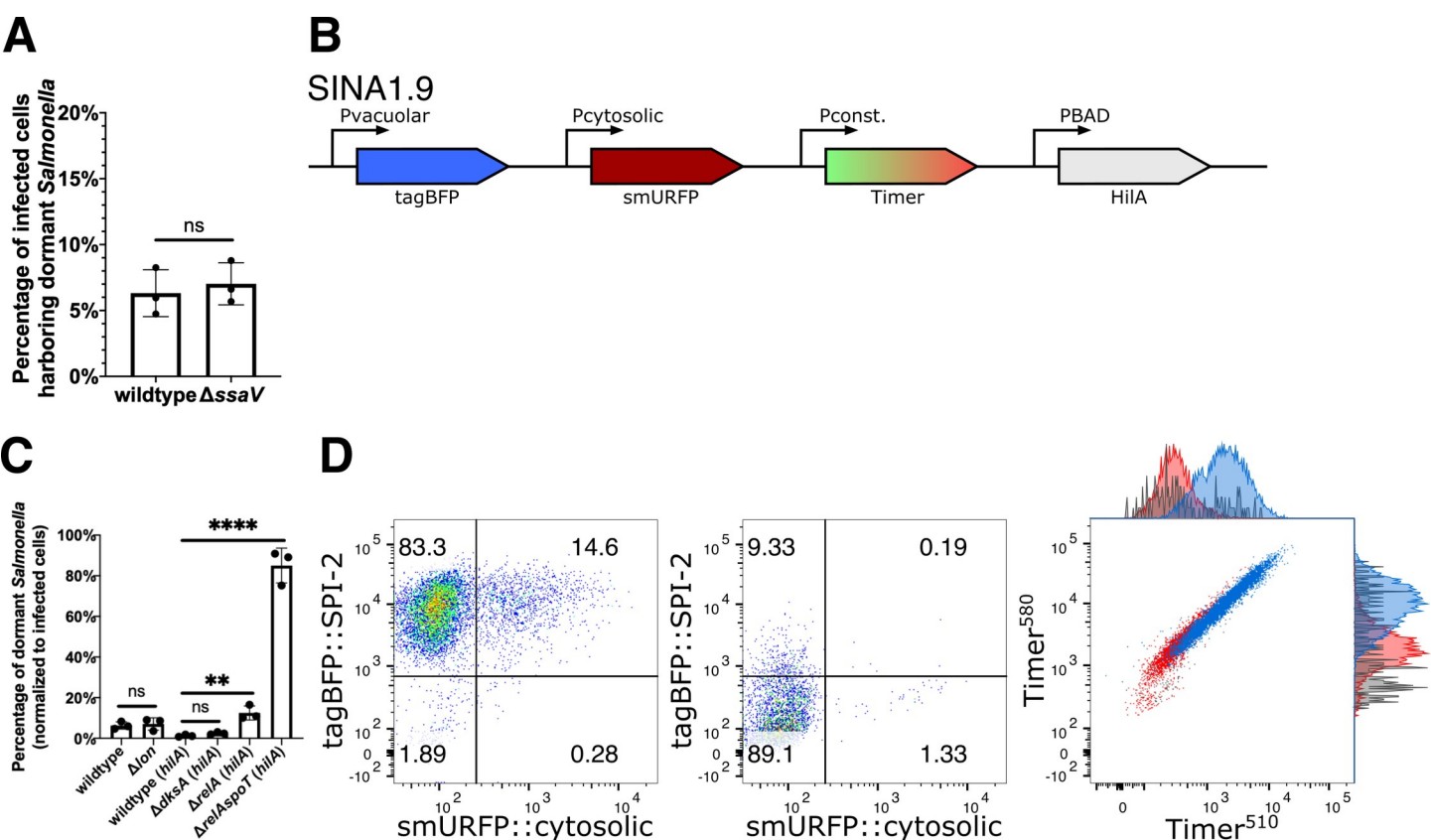

**Fig 3. *S.* Typhimurium dormancy is negatively regulated by SpoT.** (A) HeLa cells were infected with SINA1.1-harboring *S.* Typhimurium, the abundance of Vac⁻Cyt⁻ population in wild type and SPI-2 mutant *ΔssaV* infected cells were quantified with flow cytometry at 6 h pi. (n = 3) (B) Schematic diagram for the construction of SINA derivative, SINA1.9, yielded from the introduction of an arabinose-inducible *hilA* expression cassette into SINA1.1. SINA1.9 was used to rescue the reduced invasiveness of *ΔdksA*, *ΔrelA* and *ΔrelAspoT* mutant strains. (C) HeLa cells were infected with SINA1.1 or SINA1.9-harboring *S.* Typhimurium, the abundance of Vac⁻Cyt⁻ population in (p)ppGpp biogenesis and regulon mutants, *Δlon*, *ΔdksA*, *ΔrelA* and *ΔrelAspoT* were quantified by flow cytometry at 6 h pi (n = 3) (D) Distribution of Vac⁻Cyt⁻, Vac⁺Cyt⁻ and Vac⁺Cyt⁺ populations in *hilA*-expressing wild type (Left) and *ΔrelAspoT* mutant (Middle) infected HeLa cells at 6 h pi quantified by flow cytometry. Overlay Timer^bac profile (Right) of Vac⁻Cyt⁻ (red) and Vac⁺Cyt⁻ (blue) populations of wild type and Vac⁻Cyt⁻ population of *ΔrelAspoT* mutant (grey) in infected HeLa cells quantified by flow cytometry at 6 h pi. (3 independent experiments) At least a total of 1000 events of infected cells were analyzed by flow cytometry in triplicate experiments. Statistics were performed using unpaired t test. ns: not significant ($P > 0.05$), **$P < 0.01$, ****$P < 0.0001$.

synthases RelA and bifunctional (p)ppGpp synthases SpoT, after which (p)ppGpp binds to DksA to mediate transcription reprogramming for bacterial adaptation. The surge in (p)ppGpp levels also activates the ATP-dependent Lon protease to degrade Type II antitoxins to release the free toxins [36–38]. In recent reports, stringent response has been associated with slow growing *S.* Typhimurium populations, and TA systems are implicated in *S.* Typhimurium dormancy in macrophages [14,16]. Therefore, we assessed the links between the stringent response and *S.* Typhimurium dormancy in epithelial cells, studying the mutant strains (i) *ΔrelA* ((p)ppGpp synthase), (ii) *ΔrelAΔspoT* ((p)ppGpp synthases), (iii) *ΔdksA* ((p)ppGpp-binding transcription regulator) and (iv) *Δlon* (protease targeting antitoxin). With the *Δlon* mutant, we did not observe any difference in the level of dormant *S.* Typhimurium population in infected cells (~5–10%), suggesting that Lon protease is dispensable for *S.* Typhimurium dormancy in epithelial cells (Fig 3C). As *ΔrelAspoT* and *ΔdksA* were reported to suffer reduced invasiveness in epithelial cells due to the reduced SPI-1 expression, we thus constructed SINA1.9 (Fig 3B), a derivative of SINA1.1 with an additional cassette for an inducible expression of *hilA* to compensate the reduced invasiveness of the mutants (S12 Fig) following a

previously published experimental strategy [39]. With the SINA1.9-complemented mutant strains, we obtained rescued invasiveness as compared to wild type *S*. Typhimurium. This allowed us to address the requirement of (p)ppGpp biogenesis and (p)ppGpp-regulated transcription for *S*. Typhimurium persistence. A significant increase in the Vac⁻Cyt⁻ population was observed in *ΔrelAspoT*, whereas the increment was less pronounced in the *ΔrelA* single mutant and was indifferent in *ΔdksA* mutant, when comparing with the wild type strain (Fig 3C and 3D, Left and Middle panel). As SpoT has been reported to regulate SPI-2 expression [40], we further confirmed that the Vac⁻Cyt⁻ population of *ΔrelAspoT* shared a comparable metabolic profile as the one observed in the wild type strain (Fig 3D, Right panel). Together, these results suggested that (p)ppGpp stringent response mediated by SpoT but not RelA is required to restrict dormancy entry of *S*. Typhimurium within epithelial cells independent of the DksA regulon, while SPI-2 effector expression and secretion and Lon protease are dispensable.

## Discussion

*S*. Typhimurium has been reported to survive in different host cells by adopting distinctive metabolic profiles, subcellular localizations and replication rates, which has also been proposed to account for various clinical complications. Herein, we report a dormant population of *S*. Typhimurium residing in a unique vesicular compartment in epithelial cells of the intestine. These dormant epithelial *S*. Typhimurium persist within host cells for a prolonged period. The SINA reporter system was instrumental for the discovery of enterocyte-borne dormant *S*. Typhimurium as it allowed the simultaneous depiction of the metabolism, subcellular localization and replication rate of the intracellular bacteria. The compatibility of the SINA system with microscopy and flow cytometry offers the opportunity for multi-omics analysis as well as high-throughput genetic and chemical screenings on genes and compounds that influence the bacterial pathophysiology.

The dormant *S*. Typhimurium, while remaining viable in the absence of SPI-2 expression, reside in a unique vesicular compartment distinct from the RAB5, RAB7, RAB11 or LAMP1-labelled SCV or the LC3-positive autophagosomes. Upon endocytosis, endosomes are either recycled or matured and eventually degraded via fusion with lysosomes. The SCV shares such a fate if T3SS2 effectors are not secreted to hijack the vesicular maturation pathway [34]. Therefore, we propose that dormant *S*. Typhimurium reside in a vesicular compartment idle to endocytic trafficking pathways that are independent of T3SS2 effectors. Such a diversion from default endocytic pathways is also observed by other bacterial pathogens, such as *Shigella*, and despite decades of research, no strong molecular markers have been identified for the short-lived *Shigella* containing vacuole [41]. The subsequent resumption of metabolism and SPI-2 expression potentially serve as a signal to reengage the dormant membrane-enclosed *S*. Typhimurium with endocytic trafficking pathways for remodeling the SCV into the conventional replicative niche (Fig 4A). The persistence of dormant *S*. Typhimurium in host cell for up to at least 7 days in our tested condition is also striking as the bacteria inside this vacuolar compartment are likely to have restricted access to the extracellular nutrients.

*S*. Typhimurium has been reported previously to enter dormant or persistent states within a modified SCV from a range of host cell types, including macrophages and fibroblasts [12,13]. Considering the presumably identical *S*. Typhimurium dormancy observed across the different cell models, there are substantial distinctions among the targeted host cell types in terms of the detection approaches and bacterial physiology. The first *S*. Typhimurium antibiotics-tolerant persisters were identified in macrophages using a dilution reporter on non-replicating *S*. Typhimurium, which enters dormancy and a viable-but-not-cultivable state upon entry [13].

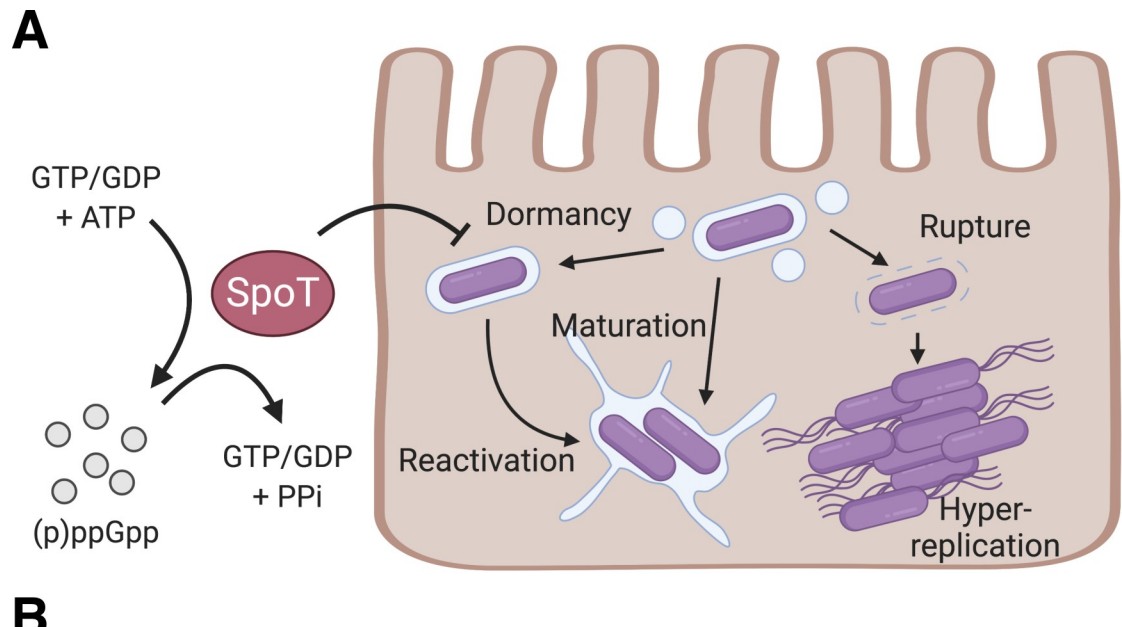

**Fig 4. Schematic illustration of the role of (p)ppGpp alarmone pathway on *S.* Typhimurium dormancy in enterocytes and the proposed pathophysiological implication of *S.* Typhimurium dormancy in enterocytes.** (A) Schematic diagram of *S.* Typhimurium lifestyles and the regulatory role of SpoT on *S.* Typhimurium dormancy in human epithelial cells. *S.* Typhimurium can opt for three distinct lifestyles: cytosolic, vacuolar and dormant, which exhibits discernible subcellular localization, replication rate and metabolism. The entry of dormant state is negatively regulated by (p)ppGpp synthatase SpoT, while the regulatory mechanism on the dormancy exit remains to be determined. (B) Schematic diagram of *S.* Typhimurium infection progression in the gut epithelium. As *S.* Typhimurium reaches the intestinal epithelium, a portion of *S.* Typhimurium expresses T3SS1 (purple) to enter host cells and adopts various intracellular lifestyles. Distinct *S.* Typhimurium lifestyles support rapid tissue colonization and gut inflammation to increase competitiveness of luminal *S.* Typhimurium (red). (Top) Reactivation of dormant *S.* Typhimurium leads to prolonged gut inflammation that supports the continuous growth of *S.* Typhimurium at gut lumen. (Bottom) Dormant *S.* Typhimurium reactivates after the eradication of gut *S.* Typhimurium, which serves as the reservoirs of infection relapse.

The dormancy is regulated by the TA system toxin, TacT that halts protein translation and induces antibiotic persistence, where the *S*. Typhimurium subsequently exits dormancy and activates SPI-2 [16,29]. Slow growing *S*. Typhimurium were also identified in fibroblasts as early as 2003, where 0.001% of the bacteria survived until the end of the studied time course [12]. Studies in fibroblasts have deciphered major genetic determinants of bacterial persistence, however the underlying mechanisms of persistence with regards to bacterial viability, antibiotics, the pathological implication have been characterized in more depth in macrophages recently [11,42]. In the different infected host cells intracellular *S*. Typhimurium activates SPI-2 to remodel the SCV for replication and interacts with endocytic trafficking pathways. In fibroblasts, the SCV subsequently interacts with host aggrephagy, where the majority of the *S*. Typhimurium residing in the SCV are eradicated whereas the remaining *S*. Typhimurium were proposed to persist [11]. In epithelial cells, the reported *S*. Typhimurium dormancy by us is distinct from that in fibroblast and macrophage. They are different in the commence of dormancy, the capacity to replicate and the SCV microenvironment [13]. The distinct niches of dormant *S*. Typhimurium may reflect cell-type specific vesicular trafficking, for example SPI-2 expression level and SCV maturation in these different target cells is not identical (S4, S8 and S9 Figs). It could also be possible that the way of entry impacts the development of dormant *S*. Typhimurium. Epithelial dormant *S*. Typhimurium is independent of Lon protease and is negatively regulated by SpoT, contrasting to that in macrophage that requires Lon protease, (p)ppGpp synthases RelA and SpoT [14,38]. The substantial difference between the *S*. Typhimurium dormancy sheds light on their potentially diverge pathophysiological implications as well as the molecular cue and mechanism that signal the establishment and exit of dormancy. The extensive work on TA systems and *S*. Typhimurium physiology in fibroblasts would serve as significant groundwork for the further studies of enterocyte-borne dormant *S*. Typhimurium [42]. It will be interesting to investigate why these regulatory modules are differentially involved in the formation of dormant *S*. Typhimurium in the different cell types, and whether the distinct niche impacts their expression and implication.

(p)ppGpp, is a bacterial alarmone that functions as a key regulator of bacterial physiology. The (p)ppGpp-mediated stringent response has been closely associated with antibiotics persistence via inhibition of protein synthesis and transcription reprogramming [43–45]. The persistent *S*. Typhimurium in macrophages is dependent on a (p)ppGpp-Lon protease-Class II TA systems axis, where TacT leads to a halt in protein translation [16]. In non-pathogenic *E. coli* and *S*. Typhimurium models, the loss of Lon and its downstream regulated TA systems leads to a diminished antibiotics persister formation due to the inactivity of toxins [46]. In our findings, the dormant phenotype is negatively regulated by (p)ppGpp synthases SpoT and partially by RelA, but independent of DksA and Lon protease-mediated pathways (Fig 4A). Our finding contrasts the current understanding on the role of stringent response on bacterial persistence, where stringent response is activated by various stress signals and (p)ppGpp synthesis would act on its molecular target to achieve persistence. Therefore, we suggest that bifunctional SpoT is required while monofunctional RelA is dispensable in *S*. Typhimurium dormancy in enterocytes, which echoes the previous report on the requirement of SpoT but not RelA in *S*. Typhimurium invasion and colonization of an *in vivo* model [39]. The essence of SpoT but not RelA for *S*. Typhimurium dormancy could suggest that either or both the (p)ppGpp hydrolysis and synthase function is required, or *rel*A is not expressed during the course of infection. Considering that *S*. Typhimurium dormancy is independent of DksA and Lon, it implies that dormancy is likely to be mediated by pathways independent of DksA transcription reprogramming and Lon protease-mediated degradation. As RelA and SpoT function to convert GDP and GTP to (p)ppGpp, and SpoT hydrolyzes (p)ppGpp to give GTP/GDP and pyrophosphate, an imbalance of RelA/SpoT activity upsets the bacterial energy status, which could

potentially act as a cue for dormancy. As dormant *S.* Typhimurium do not co-exist with *S.* Typhimurium of other lifestyles (Fig 2F), the host cell status is also implied to serve a regulatory role on *S.* Typhimurium dormancy.

With the traits we uncovered in the dormant *S.* Typhimurium within epithelial cells, this population could represent the intestinal persister, given the close association between bacterial dormancy and antibiotic persistence. Besides the proposed antibiotic persistence and horizontal gene transfer, the physiological features of enterocyte-borne dormant *S.* Typhimurium could also provide two plausible benefits to *S.* Typhimurium colonization of the host gut [47] (Fig 4B): 1) Dormancy and delayed expression of SPI-2 allow *S.* Typhimurium to evade cellular immunity during early invasion and to provide a sustained and extended SPI-2 expression at tissue scale, where *S.* Typhimurium reactivated from dormancy supports SPI-2 expression as classic vacuolar *S.* Typhimurium is eradicated. The sustained SPI-2 expression fuels gut inflammation to release electron acceptors for *S.* Typhimurium survival benefits in the gut lumen [33]. 2) Persistent *S.* Typhimurium resided within the intestinal tissue serves as the source of subsequent infection relapse or systemic spread, where the maximum duration of persistence and molecular cues for reactivation remain to be elucidated (Fig 4B).

## Materials and methods

### Mammalian cell culture

HeLa cervical adenocarcinoma cells, Caco-2 colorectal adenocarcinoma cells, 3T3 mouse fibroblasts and THP-1 acute monocytic leukemia cells were purchased from American Type Culture Collection (ATCC) and used within 20 passages of receipt. HeLa cells and 3T3 cells were cultured in Dulbecco's Modified Eagle Medium (DMEM, high glucose, GlutaMAX Supplement, ThermoFisher) containing 10% (v/v) heat-inactivated fetal bovine serum (FBS, Sigma) and incubated at 37˚C with 5% $CO_2$ and 100% humidity. Caco-2 cells were cultured in DMEM containing 10% FBS, 1% Non-essential amino acids (Gibco), 1% HEPES (Gibco), 1% Penicillin/Streptomycin (Gibco) and incubated at 37˚C with 5% $CO_2$ and 100% humidity. THP-1 cells were cultured in RPMI-1640 medium (ThermoFisher) containing 10% FBS and incubated at 37˚C with 5% $CO_2$ and 100% humidity. HeLa and 3T3 cells were seeded in 12-well tissue-culture treated plates (Corning Costar) at a density of $9x10^4$ cells/well 48 hours prior to infection. Caco-2 cells were polarized using Corning BioCoat Assay System (Corning) following manufacturer's protocol. THP-1 cells were seeded in 12-well tissue-culture treated plates at a density of $9x10^4$ cells/well 96 hours prior to infection, and differentiated in 50 μg/mL phorbol 12-myristate 13-acetate (PMA, Sigma) for 24 hours, and incubated in RPMI-1640 + 10% for 72 hours. For immunofluorescence staining, HeLa cells were seeded on UV-treated glass coverslips (Marienfeld) in 12-well plates 48 hours prior to infection. For cell sorting experiments, HeLa cells were seeded in 10 cm tissue-culture treated dishes (Corning Costar) at a density of $1.8x10^6$ cells/well 48 hours prior to infection.

### Bacterial strains

Bacterial strains and plasmids used in this study are listed in S1 and S2 Tables, respectively. All mutants were constructed using bacteriophage λ red recombinase system from parental stain *S.* Typhimurium Typhimurium strain SL1344 using primers listed in S3 Table [48]. HA-tagged T3SS2 effector strains were generated by transducing JL129 with P22 phage lysate (a generous gift from Stéphane Méresse, Centre d'Immunologie de Marseille-Luminy, France). Bacteria were cultured in Lysogeny broth (LB) supplemented with appropriate antibiotics, where necessary (Ampicillin 100 μg/mL; Kanamycin 50 μg/mL).

## Plasmid construction

The replication rate module, *Timer*[bac] is a generous gift from Dr. Dirk Bumann (University of Basel, Switzerland) [20]. To construct the localization module, *tagBFP* was amplified from pHRdSV40-NLS-dCas9-24xGCN4_v4-NLS-P2A-BFP-dWPRE using primers tagBFP_fw and tagBFP_rv, and replaced the GFP in pM973 to yield vacuolar module (pP$_{ssaG}$-*tagBFP*) [19,49]. For the cytosolic module (pP$_{uhpT}$-*smURFP*), *smURFP-HO-1* and *uhpT* promoters were amplified from pBAD *smURFP-HO-1* (smURFP_fw and smURFP_rv) and *S.* Typhimurium gDNA (uhpT_fw and uhpT_rv), respectively, and replaced the *sfGFP* and *mxiE* promoter in pTSAR1 [50,51]. The vacuolar (Vac_fw and tagBFP_rv) and cytosolic (uhpT_fw and Cyt_rv) modules were amplified and inserted into *Eco*RV and *Sma*I sites, respectively, of pBlueScript II KS (+) to generate pSINA-int. The localization module on pSINA-int was excised and inserted between *Sal*I and *Sph*I sites of pBR322 Timer[bac] to yield pSINA1.1. pSINA1.4 was generated by replacing *Timer*[bac] with *GFP* in pBR322 Timer[bac] and inserted the vacuolar module at the *Sal*I and *Sph*I sites. pSINA1.5 was generated by inserting the amplified inducible *smURFP* cassette (Ara_fw and Ara_rv) and vacuolar cassette between the *Sal*I and *Eag*I sites of pBR322 Timer[bac]. pSINA1.7 was constructed by reverting *Timer*[bac] to *DsRed* by site-directed mutagenesis using DsRed_fw and DsRed_rv. pBAD *hilA* was generated by inserting the amplified *hilA* (hilA_fw1 and hilA_rv1) between the *Bam*HI and *Pme*I sites of pBAD *smURFP-HO-1*. The inducible *hilA* cassette was amplified using primer hilA_fw2 and hilA_rv2 and inserted into the *Eco*RV site of pSINA1.1 to generate pSINA1.9.

## Bacterial infections

Bacteria strains were streaked from glycerol stock on LB agar plates with appropriate antibiotics 2 days prior to infection. Three bacterial colonies were picked for overnight culture in LB medium supplemented with 0.3 M NaCl with shaking at 37˚C. 150 μL overnight culture was subculture in 3 mL LB + 0.3 M NaCl (1:20 dilution) with shaking at 37˚C for 3 h. For strains harboring pSINA1.9, 0.1% L-arabinose was supplemented to the subculture 1 h before harvest. Bacteria were harvested with centrifugation (1 mL, 6000 x *g*, 1 min, RT), washed once in 1 x PBS and resuspended in DMEM with no FBS. HeLa cells were infected at a MOI of ~100 for 25 min at 37˚C. Extracellular bacteria were removed and washed with 1 x PBS (5X). Cells were then incubated in DMEM + 10% FBS for 1 h, washed with 1 x PBS (3X), incubated in DMEM + 10% FBS for 2 h, washed with 1 x PBS (3X) and then incubated in DMEM + 10% FBS supplemented with 10 μg/mL gentamicin for the remaining time course of the infection.

## Flow cytometry

At designated time points, cells were washed with 1 x PBS (1X) and detached with 0.05% Trypsin for 5 min at 37˚C. Detached cells were mixed with equal volume of DMEM + 10% FBS, passed through 40 μm strainer and collected by centrifugation (500 x *g*, 5 min, 4˚C). Cell pellets were dislodged and fixed in 4% PFA (15 min, RT). Fixed cells were washed with 1 x PBS (2X) and resuspended in 200 μL 1 x PBS for further analysis. For digitonin permeabilization experiment, cells were permeabilized with 45 μg/mL digitonin (1 min, RT) or 0.25% saponin (30 min, RT), then washed and stained with anti-*S.* Typhimurium primary antibody and Alexa488-conjugated goat anti-rabbit secondary antibody [21]. The fluorescence intensities of the samples were assayed with LSR Fortessa (BD) (tagBFP Ex: 405 nm Em: 450/50 nm; Timer[510] Ex: 488 nm Em: 525/50 nm; Timer[580] Ex: 562 nm Em: 582/15 nm; smURFP Ex: 633 nm Em: 670/30 nm) and analyzed with FlowJo (v10.0.4). The recorded events were gated according to the strategy described (S2 Fig).

## Cell sorting

At designated time points, cells were washed with 1 x PBS (1X) and detached with 0.05% Trypsin for 5 min at 37˚C. Detached cells were mixed with equal volume of DMEM + 10% FBS, passed through 40 μm strainer and collected by centrifugation (500 x $g$, 5 min, 4˚C). Cells were washed with 1 x PBS (1X) and resuspended in DMEM + 10% FBS supplemented with 10 μg/mL gentamicin. SYTOX Green (ThermoFisher) was supplemented to differentiate dead cells when necessary. Cells were sorted with Aria III (BD) (tagBFP Ex: 405 nm Em: 450/50 nm; Timer[510] and SYTOX Green Ex: 488 nm Em: 530/30 nm; Timer[580] Ex: 561 nm Em: 586/15 nm; smURFP Ex: 633 nm Em: 660/20 nm) to collect uninfected cells, infected cells with dormant or SPI-2 *S*. Typhimurium populations. The recorded events were gated according to the strategy described (S2 Fig).

## Immunofluorescence microscopy

Cells seeded on coverslips were washed with 1 x PBS (1X) and fixed in 4% PFA (8 min, RT). After washing with 1 x PBS (3X), cells were permeabilized and blocked in 1 x PBS, 20% FBS, 0.25% saponin (30 min, RT). Coverslips were washed with 1 x PBS (3X) and incubated with anti-RAB5, anti-RAB7, anti-RAB11, anti-LC3 or anti-LAMP1 primary antibodies and phalloidin-rhodamine diluted in 1 x PBS, 2% FBS (60 min, RT), and then washed with 1 x PBS (3X) and incubated with Cy5-conjugated goat anti-rabbit secondary antibodies diluted in 1 x PBS, 2% FBS (60 min, RT). Stained coverslips were then washed with 1 x PBS (3X) and mounted on SuperFrost Plus microscope sides (Thermo Scientific) with ProLong Gold Antifade Mountant without DAPI (Invitrogen). Samples were imaged with Perkin Elmer Ultraview confocal spinning disk microscope equipped with Volocity software and a 20X/1.3 NA air objective. Images were analyzed with FIJI (NIH) [52] and figures were prepared using Inkscape (v1.0.1).

## Colony forming unit plating

Infected HeLa cells were enriched by cell sorting, where 1000 infected cells were sorted for each sample. The cells were then collected by centrifugation at 500 x $g$ for 5 min, and subsequently lysed in 0.1% Triton X-100 for 5 min at room temperature. The lysed cells were then serially diluted and plated on LB agar plates with appropriate antibiotics.

## Dormant *S*. Typhimurium persistence assay

Infected HeLa cells harboring dormant *S*. Typhimurium were enriched by cell sorting using the gate Vac⁻Cyt⁻, and plated on 12-wells plates in DMEM + 10% FBS + Gen[10]. The medium was replaced with fresh DMEM + 10% FBS + Gen[10] to avoid the growth of *S*. Typhimurium being released from dead cells. Cells were harvested at 24 h and 168 h pi for analysis and CFU plating.

## Ciprofloxacin survival assay

A final concentration of 10 μg/mL of ciprofloxacin (CIP) were supplemented to the cell culture medium of the infected cells at 3 h pi. The cells were harvested at 6 h pi for cell sorting and CFU plating. CIP was administered at 3 h pi, which offered sufficient time for the infected population to differentiate into Vac⁻Cyt⁻, Vac⁺Cyt⁻ and Vac⁺Cyt⁺ for downstream enrichment of Vac⁻Cyt⁻, Vac⁺Cyt⁻ populations.

## Serial sectioning transmission electron microscopy

At 6 h pi, infected HeLa cells harboring Vac⁻Cyt⁻ *S.* Typhimurium were harvested by cell sorting. Enriched cells were allowed to adhere on specific dishes (MatTek) pre-coated with 50 mg/ml fibronectin (Sigma) for 3 hours. For EM sample preparation, the adhered cells were fixed with 4% PFA (EMS), 2.5% glutaraldehyde (Sigma-Aldrich) in 0.2 M HEPES for 1 hour at room temperature. Fixed samples were washed with 1 x PBS for three times, and position of interest were defined by fluorescent light microscopy. For further sample preparation, the fixed cells washed three times by the addition of fresh 0.1 M Caco buffer (pH 7.2) and post-fixed in 1% osmium (EMS) in 0.1 M Caco buffer (pH 7.2) enriched with 1.5% potassium ferrocyanide (Sigma-Aldrich) for 1 h. After three washes in 0.1M Caco buffer, samples were incubated in 0.2% of Tannic acid in water for 30 min at room temperature. Samples are post-fixed for a second time in 1% of osmium for 1 h, washed with water and incubated in 2% uranyl acetate dissolved in 25% ethanol for 1 h. Samples were then gradually dehydrated in an ethanol (Sigma-Aldrich) series ranging from 50% to 100%. Samples were embedded in PolyBed812 resin (EMS), followed by polymerization for 48 h at 60˚C. The resin embedded samples were removed from the dishes after gentle heating.

For the correlative microscopy, the region of interest was determined thanks to landmarks printed below embedded samples. Embedded cells were sectioned with an ultramicrotome (Leica, UC7) with 70 nm thickness. Thin serial sections were collected on a single slot grid (Agar scientific). Serial sections were observed with a transmission electron microscope TEM Technai T12 (ThermoFisher) at 120kV.

## Statistical analysis

Unless further specified in the figure legend, data were analyzed for statistical significance with a Mann-Whitney test using Prism 8.0 (GraphPad). $P$ value of $\leq 0.05$ is considered statistically significant. $^{*}P < 0.05$, $^{**}P < 0.01$, $^{***}P < 0.001$, $^{****}P < 0.0001$, ns: not significant/ $P \geq 0.05$.

## Supporting information

**S1 Table. *S.* Typhimurium strains used in this study.**
(DOCX)

**S2 Table. Plasmids used in this study.**
(DOCX)

**S3 Table. Primers used for molecular cloning in this study.**
(DOCX)

**S4 Table. Antibodies used in this study.**
(DOCX)

**S1 Fig. Construction strategy of SINA1.1.** The vacuolar and cytosolic modules were first individually tested with GFP (pM973 and puhpT-GFP), and then switched to tagBFP and smURFP, respectively. The vacuolar ($P_{ssaG}$-tagBFP) and cytosolic ($P_{uhpT}$-smURFP) modules were subsequently amplified and introduced into pBR322 Timer$^{bac}$ between *Sph*I and *Sal*I sites to yield SINA1.1.
(TIF)

**S2 Fig. Gating strategy of SINA1.1 reporter system.** Analyzed events were first gated for "Cells" on SSC-A vs FSC-A plot to remove cell debris. In the "Cells" events, "Uninfected" population was gated by double-negative; "Infected" was gated by double-positive on Timer$^{580}$ vs

Timer[510] plot. To gate for the basal intensity of SINA1.1 at 1 h pi, four quadrants were drawn in the "Infected" events on tagBFP::SPI-2 vs smURFP::cytosolic plot, where the biological interpretations of the four quadrants were denoted in the bottom-right sketch.
(TIF)

**S3 Fig. Localization modules indicate subcellular localization of *S*. Typhimurium.** (A) Gating strategy for applying SINA1.7 for digitonin assay. HeLa cells were infected with SINA1.7--harboring wild type *S*. Typhimurium, and harvested at 6 h pi for analysis by flow cytometry. The events were first gated for "Cells" to remove cell debris and subsequently gated for "uninfected" and "infected" based on DsRed signal. The "infected" events were subsequently gated for Vac[-]Cyt[-], Vac[+]Cyt[-] and Vac[+]Cyt[+] on tagBFP::SPI-2 vs smURFP::cytosolic plot. The fluorescence profiles FITC::*S*. Typhimurium (after immunostaining using anti-*S*. Typhimurium antibody) of Vac[-]Cyt[-], Vac[+]Cyt[-] and Vac[+]Cyt[+] and "uninfected" were plotted as overlay histograms. The gating strategy displays a positive control sample treated with saponin. (B) Schematic diagram for the constructions of the SINA derivative SINA1.7, where Timer[bac] was replaced with DsRed as compared to SINA1.1. (C) Digitonin assay on SINA-1.7 harboring wild type *S*. Typhimurium-infected HeLa cells at 6 h pi, signal intensities of uninfected (black), Vac[-]Cyt[-] (red), Vac[+]Cyt[-] (blue) and Vac[+]Cyt[+] (green) populations immunostained against anti-*S*. Typhimurium. (D) Digitonin assay on SINA-1.7 harboring wild type *S*. Typhimurium infected HeLa cells at 6 h pi, signal intensity of Vac[-]Cyt[-] population unpermeabilized (black, negative control), permeabilized with digitonin (red) and saponin (maroon, positive control). (E) Digitonin assay on SINA-1.7 harboring wild type *S*. Typhimurium infected HeLa cells at 6 h pi, signal intensity of Vac[+]Cyt[-] population unpermeabilized (black, negative control), permeabilized with digitonin (blue) and saponin (navy, positive control). (F) Digitonin assay on SINA-1.7 harboring wild type *S*. Typhimurium infected HeLa cells at 6 h pi, signal intensity of Vac[+]Cyt[+] population unpermeabilized (black, negative control), permeabilized with digitonin (green) and saponin (dark Green, positive control).
(TIF)

**S4 Fig. SINA1.1 performance in HeLa cells at 2 h, 4 h and 6 h pi.** HeLa cells were infected with wild type *S*. Typhimurium harboring SINA1.1. (A) Infected cells were harvested and analyzed at time intervals of 2 h, 4 h and 6 h pi. (Left) Timer[bac] profile of total cells at 2 h (top), 4 h (middle) and 6 h (bottom) pi in HeLa cells. (Right) Fluorescence output of the localization module of infected cells at 2 h (top), 4 h (middle) and 6 h (bottom) pi in HeLa cells. (B-C) Time-lapse microscopic acquisition of the *S*. Typhimurium intracellular lifestyle. Representative images of SINA1.1 signal output of vacuolar (B) and cytosolic (C) *S*. Typhimurium. Scare bars are 10 μm.
(TIF)

**S5 Fig. *S*. Typhimurium exhibits distinct replication rates and metabolism in HeLa cells.** HeLa cells were infected with SINA1.1-harboring *S*. Typhimurium and harvested at 6 h pi for analysis by flow cytometry. The three infected cell populations, Vac[-]Cyt[-], Vac[+]Cyt[-] and Vac[+-]Cyt[+] on tagBFP::SPI-2 vs smURFP::cytosolic plot were backgated on Timer[580] vs Timer[510] plot. Timer[580] and Timer[510] intensities were extracted from each event. (A) Quantification of Green:red ratio of Vac[-]Cyt[-], Vac[+]Cyt[-] and Vac[+]Cyt[+] population in Timer[bac] plot at 6 h pi. Green:red ratios were calculated by dividing Timer[510] by Timer[580] values, and plotted against infected cell populations. (B) Quantification of the slope of the best-fitted line of Vac[-]Cyt[-], Vac[+]Cyt[-] and Vac[+]Cyt[+] population in Timer[bac] plot at 6 h pi. For each population, a best-fitted line was plotted on the Timer[580] vs Timer[510] plot to extract the slopes for each infected cell populations. At least a total of 1000 events of infected cells were analyzed by flow cytometry in

triplicate experiments. The bars represent the mean value, unpaired t-tests were carried out, $^*P < 0.05$, $^{****}P < 0.0001$, ns: not significant.
(TIF)

**S6 Fig. Dormant *S*. Typhimurium are observed as early as 2 h pi in HeLa cells.** HeLa cells were infected with SINA1.1-harboring *S*. Typhimurium, and harvested at 1 h, 2 h and 3 h pi for analysis by flow cytometry. The infected cells were gated and the fluorescence profiles of vacuolar submodule $P_{ssaG}$-tagBFP at 1 h (black), 2 h (red) and 3 h (blue) pi were plotted as overlaying histograms.
(TIF)

**S7 Fig. Performance of SINA1.1 in Caco-2 cells.** Polarized Caco-2 monolayers were infected with SINA1.1-harboring *S*. Typhimurium and harvested at 1 h and 6 h pi for analysis by flow cytometry. (Left) Timer$^{bac}$ profile of Vac⁻Cyt⁻ (red) and Vac⁺Cyt⁻ (blue) populations and total cells (black) at 1 h (top) and 6 h (bottom) pi in Caco-2 cells. (Right) Distribution of Vac⁻Cyt⁻ and Vac⁺Cyt⁻ populations at 1 h (top) and 6 h (bottom) pi in polarized Caco-2 cells.
(TIF)

**S8 Fig. Performance of SINA1.1 in 3T3 cells.** 3T3 cells were infected with SINA1.1-harboring *S*. Typhimurium and harvested at 1 h, 6 h and 24 h pi for analysis by flow cytometry. (Left) Timer$^{bac}$ profile of Vac⁻Cyt⁻ (red), Vac⁺Cyt⁻ (blue) and Vac+Cyt+ (green) populations and total cells (black) at 1 h (top), 6 h (middle) and 24 h (bottom) pi in 3T3 cells. (Right) Distribution of Vac⁻Cyt⁻, Vac⁺Cyt⁺ and Vac⁺Cyt⁺ populations at 1 h (top), 6h (middle) and 24 h (bottom) pi in 3T3 cells.
(TIF)

**S9 Fig. Performance of SINA1.1 in THP-1 cells.** Differentiated THP-1 cells were infected with SINA1.1-harboring *S*. Typhimurium and harvested at 1 h, 6 h and 24 h pi for analysis by flow cytometry. (Left) Timer$^{bac}$ profile of Vac⁻Cyt⁻ (red) and Vac⁺Cyt⁻ (blue) populations and total cells (black) at 1 h (top), 6 h (middle) and 24 h (bottom) pi in THP-1 cells. (Right) Distribution of Vac⁻Cyt⁻ and Vac⁺Cyt⁻ populations at 1 h (top), 6 h (middle) and 24 h (bottom) pi in differentiated THP-1 cells.
(TIF)

**S10 Fig. Serial sectioning TEM determines the subcellular localization of Vac⁻Cyt⁻ *S*. Typhimurium.** (A) Brightfield and fluorescent microscopy image of region of interest on MatTek dish. (B) Brightfield and fluorescent microscopy image of cells of interest harboring Vac⁻⁻Cyt⁻ *S*. Typhimurium. (C) TEM image of cell of interest in labelled region from (B). (D) Magnified TEM image of labelled region from (C).
(TIF)

**S11 Fig. Infected cells harboring dormant *S*. Typhimurium are viable.** HeLa cells were infected with SINA1.7 harboring *S*. Typhimurium, harvested at 6 h pi and stained with SYTOX Green and analyzed by flow cytometry. The infected cells were gated and the fluorescence profiles of SYTOX Green in uninfected cell (black), Vac⁻Cyt⁻ (red), Vac⁺Cyt⁻ (blue) and Vac⁺Cyt⁺ (green) were plotted as offset histograms.
(TIF)

**S12 Fig. Ectopic expression of *hilA* rescues the loss of invasiveness.** HeLa cells were infected with various *S*. Typhimurium strains and harvested at 6 h pi for flow cytometry analysis. The losses of invasiveness in Δ*dksA* and Δ*relAspoT* mutants are rescued by ectopic expression of *hilA* from the arabinose inducible cassette in SINA1.9. At least a total of 1000 events of infected

cells were analyzed by flow cytometry in triplicate experiments. The bars represent the mean value, unpaired t-test was carried out, $^{****}P < 0.0001$.
(TIF)

**S1 Movie. Connected to Fig 1D: Time-lapse microscopy shows the fluorescence signal output from SINA1.1 in Vac$^+$Cyt$^-$ intracellular *S.* Typhimurium population.** Brightfield and fluorescence output of Timer$^{bac}$, P$_{ssaG}$ and P$_{uhpT}$ from SINA1.1-harboring *S.* Typhimurium exhibiting Vac$^+$Cyt$^-$ profile. Images were taken every 15 min starting from 1 h pi. (AVI).
(AVI)

**S2 Movie. Connected to Fig 1D: Time-lapse microscopy shows the fluorescence signal output from SINA1.1 in Vac$^+$Cyt$^-$ and Vac$^-$Cyt$^+$ intracellular *S.* Typhimurium population.** Brightfield and fluorescence output of Timer$^{bac}$, P$_{ssaG}$ and P$_{uhpT}$ from SINA1.1-harboring *S.* Typhimurium exhibiting Vac$^+$Cyt$^-$ and Vac$^-$Cyt$^+$ profiles. Images were taken every 15 min starting from 1 h pi. (AVI).
(AVI)

**S3 Movie. Connected to Fig 1D: Time-lapse microscopy shows the fluorescence signal output from SINA1.1 in Vac$^-$Cyt$^-$ intracellular *S.* Typhimurium population.** Brightfield and fluorescence output of Timer$^{bac}$, P$_{ssaG}$ and P$_{uhpT}$ from SINA1.1-harboring *S.* Typhimurium exhibiting Vac$^-$Cyt$^-$ profile. Images were taken every 15 min starting from 1 h pi. (AVI).
(AVI)

## Acknowledgments

We thank the members of the Dynamics of Host-Pathogen Interactions Unit for the constructive comment and discussion. We are grateful for the generous plasmid gift from D. Bumann, and we acknowledge S. Méresse for the P22 lysates of tagged PipB2-2HA.

## Author Contributions

**Conceptualization:** Chak Hon Luk, Jost Enninga.

**Data curation:** Chak Hon Luk, Camila Valenzuela, Magdalena Gil, Léa Swistak, Yuen-Yan Chang.

**Formal analysis:** Chak Hon Luk, Camila Valenzuela, Magdalena Gil, Léa Swistak, Yuen-Yan Chang, Jost Enninga.

**Funding acquisition:** Chak Hon Luk, Jost Enninga.

**Investigation:** Chak Hon Luk, Camila Valenzuela, Magdalena Gil, Léa Swistak, Yuen-Yan Chang, Adeline Mallet, Jost Enninga.

**Methodology:** Chak Hon Luk, Léa Swistak, Perrine Bomme, Adeline Mallet.

**Project administration:** Jost Enninga.

**Resources:** Jost Enninga.

**Supervision:** Jost Enninga.

**Validation:** Camila Valenzuela, Magdalena Gil, Léa Swistak.

**Visualization:** Chak Hon Luk, Perrine Bomme, Adeline Mallet, Jost Enninga.

**Writing – original draft:** Chak Hon Luk, Jost Enninga.

**Writing – review & editing:** Chak Hon Luk, Jost Enninga.

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
