## [Decision Letter · Decision Letter 0]

28 Dec 2020

Dear Dr Enninga,

Thank you very much for submitting your manuscript "Salmonella  endorses a dormant state within human epithelial cells for persistent infection" for consideration at PLOS Pathogens. As with all papers reviewed by the journal, your manuscript was reviewed by members of the editorial board and by several independent reviewers. In light of the reviews (below this email), we would like to invite the resubmission of a significantly-revised version that takes into account the reviewers' comments.

As you will see from the reviewers' comments, there was some consensual interest and enthusiasm for the development and potential of the SINA tool reported in your manuscript. However, all reviewers raised some significant concerns about the design of some experiments, consideration of the existing literature on *Salmonella *persisters, and the validity of many important conclusions. All of these concerns need to be carefully addressed in a revised manuscript.  

While many of the reviewers’ comments can be addressed with textual changes, clarifications and revisions of data presentation, addressing several of these concerns requires additional experiments to be performed.  In particular, it appears essential to address the following issues raised by the reviewers in your revised manuscript:

All reviewers requested clarifications on the expression of SPI-2 in the newly described dormant population of intracellular *Salmonella*, via methodologies that would complement and validate the vacuolar reporter part of SINA. This also includes discussing the known roles of SpoT in SPI-2 expression and persisters.Reviewers 1 and 3 requested a deeper characterization of the vacuolar compartment occupied by the dormant *Salmonella *population, as the data provided in the manuscript was deemed superficial. Reviewer 2 requested changes in the experiments testing reactivation of dormant bacteria using Ciprofloxacin and addressing the caveat of HilA overexpression on the intracellular behavior of dormant bacteria in the context of SpoT/RelA functions.Reviewer 2’s request to include an experimental comparison of your findings in the fibroblast model of *Salmonella *persisters should be addressed, given the significance of this model. 

We cannot make any decision about publication until we have seen the revised manuscript and your response to the reviewers' comments. Your revised manuscript is also likely to be sent to reviewers for further evaluation.

Sincerely,

Jean Celli

Guest Editor

PLOS Pathogens

Raphael Valdivia

Section Editor

PLOS Pathogens

Kasturi Haldar

Editor-in-Chief

PLOS Pathogens

orcid.org/0000-0001-5065-158X

Michael Malim

Editor-in-Chief

PLOS Pathogens

orcid.org/0000-0002-7699-2064

Reviewer's Responses to Questions

**Part I - Summary**

Reviewer #1: In this manuscript by Luk et al., the authors construct a new tool (SINA) to characterize distinct populations of Salmonella within enterocytes. They use SINA to identify subpopulations of intracellular Salmonella and describe a population of bacteria that are not within the “classical” SCV or within the cytosol, but seem to reside in an uncharacterized vesicular compartment in a dormant state. The dormant bacteria do not express SPI2 at 6 hours post-infection, but appear to express it at later time points. Finally, they propose that the dormant state is regulated by (p)ppGpp-stringent response through SpoT. The authors propose that dormant Salmonella within epithelial cells mediate persistence. This is a potentially interesting study, however, additional data and controls are needed. The manuscript would benefit from clearly defining at the beginning what dormancy is and how does this compare to persisters.

Reviewer #2: The study has as major strength the combination of previously used fluorescent reporters in new tool that allows to analyze simultaneosuly location and growth status of intracellular Salmonella. Apart from this technical advance, the study is conceptually weak, lacking much comparative discussion with prior work on Salmonella persisters in other cell models. Presentation of the data has also flaws, with some experiments described superficially and obvious controls missing. For example, the “colony forming unit plating” assay described in pag. 19 relates to the data shown in Fig. 4A, but in none of the two sites the cell line in which it was done, is specified. This assay involves a long-term infection (168 h, 7 days). It is not easy to imagine the experimental set-up when using epithelial cells as HeLa that are permissive for bacterial proliferation. Experimental details must be therefore precisely exposed. In general, there is clear overstatement in the conclusions due to the lack of more convincing data.

Reviewer #3: The authors use Salmonella infection as a model to address important questions about bacterial growth in different compartments of host cells. They also ask where Salmonella persisters colonize host cells, and how this is regulated. They develop a new probe (SINA) that indicates both bacterial localization and also metabolic activity at the same time. They propose that a population of dormant bacteria occupies an uncharacterized compartment and can be re-activated upon favorable growth conditions. The strength of this study is the development of new methods and they test these both in vitro and in vivo. This is an interesting study and the results are promising, but I have some questions about the probe and the interpretation of the data.

**Part II – Major Issues: Key Experiments Required for Acceptance**

Reviewer #1: 1) the authors claim that the dormant state mediates persistence, but they don’t really show this. First off, they need to define what they mean by persistence. The data presented in Figure 4 are not very convincing. They only show 6 h, 24 h, and 168 h. What happens at times in between? They see that ~50% of the “dormant” bacteria begin to express SPI2 at about 24 hours. Do these bacteria replicate, burst out and reinfect neighboring epithelial cells to start the cycle again? Showing images of these longer infections and more time points in between 1 and 7 days would help sort this out.

2) Movies and images need to show host cells with either phase of DIC imaging. For example, in movie 3, I see some red and green fluorescence, but I have no idea whether they are inside of host cells or extracellular.

3) I’d like to see better characterization of the “unique” vesicular compartment that they claim dormant Salmonella reside in. Fig.3 only shows 1 time point, 6 h and I don’t know what “minor” recruitment of LAMP1 means. Can this be quantitated over a time course? Also, transmission electron microscopy would be nice confirmation that these bacteria are really in a vesicle of some sort.

Reviewer #2: This study combines fluorescent reporters previously used to monitor subcellular localization of Salmonella (vacuolar, cytosolic) with a reporter that provides measure of replication rate based on a “Timer” dsRed variant.

The study does not have a clear starting hypothesis, but the application of this new dual reporter, named SINA, led to the authors to claim the existence of a “third” population of intracellular Salmonella that is intra-vacuolar and would remain in a dormant state form early infection times..

The first part of the study describing the new tools is technically convincing. However, the subsequent sections are much less solid and introduce concepts not supported by the data shown. Moreover, the study is not rigorous with the literature and it lacks many important controls being clearly biased by the data shown in macrophages. This phagocytic cell type is the sole comparator model used, leaving aside and omitting important observations in other models that provided valuable data on Salmonella intracellular persistence.

The study therefore forgets important observations in the Salmonella field regarding the capacity of this pathogen to establish a dormant state inside eukaryotic cells as well as the role of SpoT in controlling the intracellular virulence program. These omissions refer to the work performed in the fibroblast model, including: i) strong SPI-2 activity in non-growing intracellular Salmonella (Nunez-Hernandez et al., Infect Immun 2013, 2014) ; ii) production of several toxin-antitoxin modules by non-growing intracellular S. Typhimurium (Lobato-Marquez et al., Scientific Reports, 2015), none of which was tested here; iii) the selection of genetically table S. Typhimurium mutants with 3-4- logs higher capacity to persist inside the fibroblasts (Cano et al. Infect Immun 2003, 71: 3690-3698); and, iv) the role of SpoT, but not RelA, in producing ppGpp to stimulate activity of SPI-2 in intraphagosomal Salmonella (Fitzsimmons et al., mBio, 2020, 11:e03397-19).

The study also lacks some rational justifications on why to choose exclusively the RelA-SpoT-DskA-Lon axis despite being forced to use “artificial conditions” in which SPI-1 is boosted by in-trans expression of HilA. Invading bacteria with high SPI-1 levels might be transiently prevented to correctly induced the SPI-2 intracellular phase.

These are key experiments that should be addressed:

1) Ciprofloxacin survival assay (p. 19): The experiment involves incubation of infected cells in medium containing the drug. This design does not take into account that the bactericidal action of the drug can modify the physiology of intravacuolar bacteria to render it Vac- from a Vac+ status at the beginning of the treatment. Moreover, the Vac+Cyt+ epithelial cells contain both cytosolic bacteria undergoing rapid growth and intra-vacuolar bacteria with reduced growth rate. This Vac+Cyt+ cell population is included in the Material and Methods’ section (pag. 19) but not shown in Fig. 2D. So, it is uncertain whether or not was tested. Bactericidal effect might be different depending on the growth status inside the host cell. Furthermore, accessibility of the drug to the vacuolar compartments or the cytosol might not be similar. A more convincing experiment is to isolate bacteria from the different populations: Vac-Cyt- and Vac+Cyt- and further exposed to the drug in the test tube for 3 h and then do plating to count CFU next day.

2) Considering the absolute of biochemistry in this study, it is recommended to tag in the chromosome a SPI2-specific gene (e.g. ssrB) and evaluate relative levels in the distinct populations (Vac-Cyt-, Vac+Cyt-, Vac+Cyt+). This must be done after sorting. This approach was useful to demonstrate SPI2 activity in non-growing Salmonella inside fibroblasts (Nunez-Hernandez et al., Infect Immun 2014).

3) It is not obvious why the RelA-SpoT-DskA-Lon axis must be the only that operates to trigger generation of dormant Salmonella in epithelial cells. The work of Lobato-Marquez et al. (Sci Reports, 2015) showed that various TA systems operate in intracellular S. Typhimurium in both epithelial cells and fibroblasts. Unlike the relA, relAspoT and dskA mutants, none of the mutants lacking these TA systems were found to be impaired for invasion. So, no need to artificially manipulate SPI-1. These should be tested.

Reviewer #3: -The new SINA probe is being used to identify a population of dormant bacteria. Are there any other bona fide markers/probes that can be used in tandem to confirm what the authors are concluding? Perhaps RT-PCR or mass spec analysis of cell sorted bacteria would help?

-The new probes are plasmids. Is it stably integrated into the bacterial chromosome? If not, what evidence is there that plasmid loss is not occurring? Again, maybe RT-PCR would help.

-The authors refer to bacteria lacking expression of ssaG (SPI-2 T3SS component, used as marker of vacuolar) and uhpt (hexose 6-phosphate: phosphate antiporter; used as marker of cytosolic bacteria) and having a low metabolic activity as being “dormant”. An alternative interpretation is that these bacteria merely lost the plasmid containing the SINA probe. Or do the bacteria go to a non-acidified compartment without hexose 6-phosphate?

-Supplemental Movie 1. What is the movie supposed to be showing? It might help to have a separate figure with frames from the movie where bacterial signals and populations are highlighted.

-Figure 1D, the labeling is confusing, and not consistent with figure legend. For example, “6h SPI-2” not the same as “vacuolar”. When I first read this I thought a mutant was used, so confused. The addition of colour only to the merged channel makes the situation worse. Importantly, what is the conclusion from this figure? Are vacuolar bacteria less metabolically active than those in the cytosol? Is there a population of fast and slow growing bacteria in the cytosol?

-Figure 1E. Why are there only Vac+Cyt+ bacteria, and not Vac-Cyt+ bacteria? What does this suggest about the probe, or potentially the bacterial entry to the cytosol? The authors should discuss this point.

-pg8 “This revealed that the Vac-Cyt- S.Typhimurium exhibit a similar replication rate (S5A Fig) but a reduced metabolic activity (S5B Fig) compared to Vac+Cyt- S. Typhimurium as depicted by the green:red ratio and slope of Timerbac plot, respectively”. This is a confusing conclusion: how can the authors propose a similar replication rate with reduced metabolic activity? It seems likely the probe is providing an artefactual lack of signal, or the bacteria are in an unusual location such as the cell surface.

-are antibiotics present in the medium during experiments? The authors point out that antibiotics affect dormancy…. Is this driving observed phenotypes? What happens when gentamicin is removed from the medium?

-Movie S3. There does not appear to be any signals. What are we supposed to be looking at?

-pg9, “With SINA1.5, we observed that approximately half of the Vac-

S. Typhimurium did not respond to arabinose induction at designated time intervals

(Fig 2C). Combining this with our observations on the reduced metabolism (S5B Fig),

we were able to put forward the first evidence to propose that the Vac-Cyt- S.

Typhimurium adopts a dormant state (coined as dormant S. Typhimurium hereafter)”. Is arabinose cell permeable? Could a lack of induction simply reflect a disconnection of bacteria in unusual vacuoles from the endocytic pathway?

-pg10 “Therefore, we conclude that dormant S.Typhimurium are localized within a unique membrane-bound compartment distinct from the conventional SCV”. How can the authors conclude the bacteria are present in a cellular compartment if no markers are present? The authors should consider using correlative electron microscopy to visualize what they are calling a dormant population within cells. Staining for ubiquitinated proteins (typically seen on cytosolic bacteria) should also be considered.

-Related to above, what is the morphology of the dormant bacteria? Upon entry to the cytosol some Salmonella increase in size and decrease in antibody staining… does this type of visual analysis suggest dormant bacteria are in the cytosol?

-Are dormant Salmonella found in cells that have Salmonella-induced filaments (Sifs)? Or those that don’t? This would be interesting to know since bacteria that make Sifs are associated with growth of the bacteria.

-Figure 4. Dormant bacteria were isolated by flow sorting… can the authors perform microscopy to show what was purified? Importantly, did the collected bacteria bear the selectable marker of the SINA probe?

**Part III – Minor Issues: Editorial and Data Presentation Modifications**

Reviewer #1: 1) many sentences don’t make sense. I will give one example of many, e.g. on page 14, first paragraph “This discrepancy lies in the commence of dormancy, capacity to replicate and permeability to the surrounding microenvironment.” I believe that this manuscript would benefit from editing for grammar, word choice and sentence structure.

2) I’m not fond of the use of “endorses” in the title.

3) How does SpoT regulate dormancy? Does it repress SPI2 expression?

4) The figure legends should include all details of statistical analyses and numbers of cells and repetitions for each panel.

Reviewer #2: - Abstract (lines 14-16): In no part of the study the authors show that the dormant bacteria Vac-Cyt- express SPI-2 al later infection times. The next sentence assumes that dormant bacteria could have “lost” SPI2 activity, which is also not demonstrated by any experiment.

- Pag.3 Importance of the work: Line 4 (We found….). This reviewer does not agree with this statement. It assumes, but it does not prove, that formation of dormant bacteria in epithelial is the one that operates in macrophages. Moreover, TacT is vaguely cited, not functionally tested and no phenotypic test is done with any other TA systems reported to be induced by dormant S. Typhimurium.

- Fig. 1C. The colours expected with the two reporters are shown in two cells but the respective names of the reporters are missing.

- Figure 4A. Discussing these data the authors do not cite the work published in 2003 showing cultivability of dormant S. Typhimrium recovered from fibroblasts after one-week infection (Cano et al. Infect Immun.)

- Although the authors insist that regulation of persisters’ formation might differ among distinct cell types, no comparative analysis is shown. This is extremely relevant considering that the studies in fibroblasts as well as those of Helaine’s group in macrophages have linked SPI2 activity to dormant intracellular Salmonella. The only exception is Fig. S8 with THP1 macrophages, in which up to 70% of the Vac-Cyt- population is shown. Where are then the dormant bacteria reported by Helaine’s group present in these macrophages?

- Supplementary figure S4 clearly shows that the Vac-Cyt- population in HeLa decreases over time, chaining from 37.7% at 2 hpi to 7.41% at 6 hpi. The Vac+Cyt- population however remains rather stable, moving from 60% to 76% at 6 hpi. Authors should comment on the significance of this decrease.

- Description or discussion of some data do not reflect accurately the results shown with not alternatives being considered. An example is the “loss” of SPI2 expression in the Vac-Cyt- population, mentioned repeated times. The authors do not discuss about the possibility of these bacteria not triggering SPI2 expression at any time of the infection and remaining frozen at some early stage of SCV biogenesis.

- Another missing discussion is that related to the possibility of Vac-Cyt- intracellular bacteria being masked by other populations the same infected cell. Heterogeneity regarding epithelial cells harbouring Vac+ and Cyt+ bacteria was first characterized by Knodler’s lab and it is also evident in this study. So, Vac-Cyt- bacteria can be also present in some of the infected populations assigned as Vac+Cyt- and Vac+Cyt+.

- The data shown compare the Vac-Cyt- population in other cell lines such as Caco-2 and THP1 are surprising in the sense that they show a much larger Vac-Cyt- populations (34% and 70% at 6 hpi, respectively for each cell line). This reviewer wonders why these cell lines were not selected better than HeLa for the purpose of demonstrating the existence of such dormant bacteria. An additional cell type missing are fibroblasts, in which dormant population predominates versus proliferating bacteria. The SINA tool could be of impressible utility in these alternative models.

- Genetic background (genotype) of the S. Typhimurium wild-type strain in which plasmid were inserted should be indicated in the Table of strains, no just in the text.

- Lines 3 and 5, 3rd paragraph, page 4. Include references.

- Line 12, 3rd paragraph, page 4. Provide more details about the specific metabolic activity.

- Line 2, 1st paragraph, page. Reference of Lobato-Marquez et al. (Sci Rep, 2015) is missing when referring to TA systems.

- The manuscript was not line numbered. To be corrected in future versions.

- The distinct panels of the videos (S1-to-S3) are not labelled for the respective channel shown. Moreover, the lower panel is pixelated in grey scale with no clear image(s) seen.

- Reference 13 is incomplete. It has no journal information.

- 2nd paragraph, page 5: TacT is not introduced first time it is cited.

- Pages 7-9. Figure S3 is cited after S4 and S5.

- Fig 2C: use better contrasting colours for the bars shown in middle panel.

- Fig 2D: What are it the lines connecting the percentage of Dormant and Vac+Vac- populations? Are they independent biological replicas? The legend refers to n = 3.

- Figure 3B: it lacks staining of representative bacteria from Vac+Cyt- and Vac-Cyt+ populations. Labelling of early-medium markers (Rab5, Rab7) is also recommended to dissect the exact nature of compartment in which these dormant are.

- What is Fig. 4B? Please provide a scheme of the experimental design. The way it is described in the legend is not clear at all.

- Line 1, 4th paragraph, page 11. First sentence seems incorrect. Please, rephrase.

- Page 13. Discussion refers to “first” S. Typhimurium persisters reporter in macrophages. Chronologically, the last paragraph is not correct. See comments above about earlier studies in fibroblasts.

- Discussion is unnecessary long. It should be shortened by at least one page.

Reviewer #3: -Title: I’m not sure “endorses” is the right word to use…

-Short title: perhaps “in enterocytes” is better?

-Supplemental movies: it would help to label the frames so we know what we are looking at.

PLOS authors have the option to publish the peer review history of their article (what does this mean?). If published, this will include your full peer review and any attached files.

Reviewer #1: No

Reviewer #2: No

Reviewer #3: No
---

## [Decision Letter · Decision Letter 1]

8 Apr 2021

Dear Dr Enninga,

We are pleased to inform you that your manuscript 'Salmonella  enters a dormant state within human epithelial cells for persistent infection' has been provisionally accepted for publication in PLOS Pathogens.

Best regards,

Jean Celli

Guest Editor

PLOS Pathogens

Raphael Valdivia

Section Editor

PLOS Pathogens

Kasturi Haldar

Editor-in-Chief

PLOS Pathogens

orcid.org/0000-0001-5065-158X

Michael Malim

Editor-in-Chief

PLOS Pathogens

orcid.org/0000-0002-7699-2064

Dear Jost,

The revised version of your manuscript was sent back to the reviewers, who were satisfied with your responses to their original comments. I concur with their recommendations and consider that no further revisions are needed. Congratulations on this very interesting study.

Best regards,

Jean

Reviewer Comments (if any, and for reference):

Reviewer's Responses to Questions

**Part I - Summary**

Reviewer #1: The authors have addressed my concerns.

Reviewer #3: (No Response)

**Part II – Major Issues: Key Experiments Required for Acceptance**

Reviewer #1: (No Response)

Reviewer #3: (No Response)

**Part III – Minor Issues: Editorial and Data Presentation Modifications**

Reviewer #1: (No Response)

Reviewer #3: (No Response)

PLOS authors have the option to publish the peer review history of their article (what does this mean?). If published, this will include your full peer review and any attached files.

Reviewer #1: No

Reviewer #3: No

---

## [Editor Report · Acceptance letter]

28 Apr 2021

Dear Dr. Enninga,

We are delighted to inform you that your manuscript, "*Salmonella* enters a dormant state within human epithelial cells for persistent infection ," has been formally accepted for publication in PLOS Pathogens.

Best regards,

Kasturi Haldar

Editor-in-Chief

PLOS Pathogens

orcid.org/0000-0001-5065-158X

Michael Malim

Editor-in-Chief

PLOS Pathogens

orcid.org/0000-0002-7699-2064